# Where Computation Lives Inside TabPFN:
# Causal Localisation of Attention Head Function

**Atharva Gupta** [1] **Dhruv Kumar** [1] **Murari Mandal** [2] **Saurabh Deshpande** [3]

## Abstract

We present the first causal mechanistic analysis of a tabular foundation model, investigating how TabPFN-2.5's feature-wise attention heads distribute computation across layers. Using activation patching, ablation, and attention entropy across two synthetic regression datasets, we find clear temporal specialisation: one head's causal necessity dominates that of the others by 2 to 5 times at peak layer, with its dominant layer shifting across tasks of different complexity, while the remaining heads exhibit symmetric late layer profiles. Attention entropy and patching provide convergent evidence for the computationally active layers of the dominant head. We additionally conduct an extremely preliminary investigation of inference-time steerability via contrastive activation steering. In these initial experiments, steering fails to transfer across samples. We hypothesize that this is because TabPFN's in-context learning mechanism encodes task structure through context-dependent attention rather than the stable parametric directions that make steering tractable in language models.

## 1. Introduction

Trained on synthetic structural causal models (Peters et al., 2017; Pearl, 2009), TabPFN-2.5 (Grinsztajn et al., 2025) acquires strong predictive capabilities that transfer to new tabular tasks through in-context learning. Despite continued scaling of context size (Hollmann et al., 2025) and sustained competition from tree-based methods (Grinsztajn et al., 2022; Gorishniy et al., 2021), mechanistic understanding of how TabPFN produces its predictions has lagged behind: existing interpretability work is confined to post-hoc attribution (Grinsztajn et al., 2025; Rundel et al., 2024). A recent exception is Knauer & Rodner (2026), who find correlational evidence that individual neurons exhibit selective responses to high-level concepts, motivating a causal investigation of where and how.

Mechanistic interpretability has made progress in language models (Olsson et al., 2022; Elhage et al., 2021; Todd et al., 2024) and time-series transformers (Wiliński et al., 2025), but no comparable causal analysis exists for tabular architectures.

We address this gap with a causal mechanistic study grounded in a concrete question: **Which components of TabPFN-2.5 are causally responsible for specific computations, and at which layers do they emerge?** Using causal activation patching (Meng et al., 2023; Heimersheim & Nanda, 2024) and ablation across two synthetic datasets, we identify two functional head classes in TabPFN-2.5's feature-wise self-attention: one head whose ablation effect dominates the others on both datasets (with the peak layer differing across tasks), and two heads exhibiting symmetric patching and ablation profiles at late layers on the simpler task. We additionally investigate inference-time steerability via contrastive activation steering (Turner et al., 2023; Panickssery et al., 2023); directions do not transfer to held-out samples, a result we attribute to TabPFN's relational ICL mechanism (Appendix G). To the best of our knowledge, this is the first causal mechanistic analysis of a tabular foundation model.

## 2. Experiments and Results

### 2.1. Preliminaries

TabPFN-2.5 takes a labelled training set and a test instance as input and predicts the test label in a single forward pass via in-context learning (Hollmann et al., 2025; Grinsztajn et al., 2025). Full experimental hyperparameters are listed in Appendix A. The model applies two sequential self-attention mechanisms per layer: `self_attn_between_items` across the sample dimen-

Code available at https://github.com/atharva7-g/tabfm-interp. [1]Department of Computer Science, Birla Institute of Technology and Science, Pilani, India [2]School of Computer Engineering, Kalinga Institute of Industrial Technology, Bhubaneswar, India [3]Birla AI Labs, Office of Ananya Birla, Aditya Birla Group, India. Correspondence to: Atharva Gupta <f20240519@pilani.bits-pilani.ac.in>.

*Proceedings of the $2^{nd}$ ICML Workshop on Foundation Models for Structured Data*, Seoul, South Korea. 2026. Copyright 2026 by the author(s).

sion and `self_attn_between_features` across the feature-block dimension within each sample. We focus on `self_attn_between_features`: it is the only module that operates across feature representations, making it the natural locus for cross-feature computation in regression tasks. We use the regression model (`TabPFNRegressor`) (Grinsztajn et al., 2025; Prior Labs, 2025a), which has 18 transformer layers, $H=3$ attention heads with $d_h=64$, and $d_{\text{model}}=192$ (Prior Labs, 2025b). Per-dataset feature block counts and their derivation are in Appendix D (Table 5). The label $y_i$ is embedded as the final token. Throughout, a head index (e.g. "Head 2") refers to a distinct, separately parameterised module at each layer; unless explicitly demonstrated otherwise, we do not claim that heads sharing an index across layers form a single functional circuit.

## 2.2. Datasets

**Multiplication Dataset.** Each sample has three features $a, b, c \sim \mathcal{N}(0, 1)$ with target $y = a \cdot b + c$, combining a nonlinear interaction $(a \cdot b)$ with an additive term $(c)$. The low dimensionality $(d = 3)$ allows direct inspection of individual feature contributions.

**Pairwise-50 Dataset.** Each sample has $d = 50$ features $x_1, \ldots, x_{50} \sim \mathcal{N}(0, 1)$ with target

$$y = \sum_{i=1}^{50} \sum_{j=1}^{50} x_i \cdot x_j = \left( \sum_{i=1}^{50} x_i \right)^2.$$

The $50 \times 50 = 2{,}500$ terms cover all ordered pairs and self-products.

## 2.3. Causal Mechanistic Analysis

### 2.3.1. ACTIVATION PATCHING

Activation patching (Meng et al., 2023; Heimersheim & Nanda, 2024) tests causal responsibility by replacing a hidden activation in a corrupted forward pass with the corresponding value from a clean run, then measuring output recovery. Concretely, for a clean run $x^{\text{clean}}$ and corrupted run $x^{\text{corr}}$, patching site $S$ at layer $\ell$ replaces $h_{\ell,S}^{\text{patched}}(x^{\text{corr}}) \leftarrow h_{\ell,S}(x^{\text{clean}})$, and for a scalar regression output $\hat{y}(\cdot)$ we report the normalised restoration score

$$\text{Restore}(\ell, S) = \frac{\hat{y}(x_{\ell,S}^{\text{patched}}) - \hat{y}(x^{\text{corr}})}{\hat{y}(x^{\text{clean}}) - \hat{y}(x^{\text{corr}})},$$

i.e. "recovery" is the fraction of the clean-vs-corrupted output gap restored by the patch (0% = no restoration, 100% = full restoration; values outside $[0, 100]\%$ indicate the patch over- or counter-shoots the gap). Full setup and metrics are in Appendix B.1. We sweep over layers and intervention sites as summarized in Figure 1.

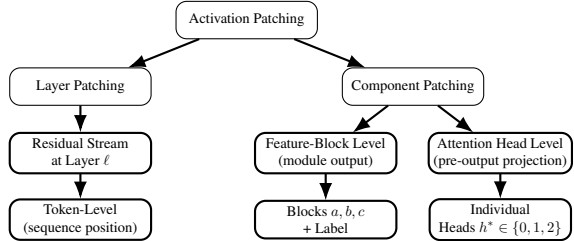

*Figure 1.* Activation patching hierarchy. Component patching targets `self_attn_between_features` at two granularities: feature-block level (post-projection output) and attention head level (per-head outputs before $W_O$; see Appendix C). Token-level patching results are in Appendix F.

Layer-level patching — replacing the full residual stream at depth $\ell$ with the clean-run value — achieves $\approx 100\%$ prediction recovery at every layer (Appendix B.1), consistent with a highly distributed representation in which information sufficient for correct prediction is preserved throughout the network.

### 2.3.2. COMPONENT-LEVEL PATCHING

The feature-wise self-attention module applies standard multi-head attention (see Appendix C) across the $\lceil (2d + 1)/3 \rceil + 1$ feature blocks per sample (see Appendix D). We focus on attention head level interventions; feature-block patching and ablation results are deferred to Appendix B.3 and B.4.

**Attention head level.** We patch individual heads before $W_O$ (the output projection; defined in Appendix C), at the point where the $H$ per-head outputs are concatenated. Let $\hat{V}_\ell \in \mathbb{R}^{B \cdot N \times F \times H \times d_h}$ denote the pre-projection per-head outputs, where $F = \lceil d'/3 \rceil + 1$.

Patching head $h^*$ replaces $\hat{V}_\ell[:, :, h^*, :] = \hat{V}_\ell^{\text{clean}}[:, :, h^*, :]$. The patched tensor is then passed through $W_O$, isolating each head's individual causal contribution.

## 2.4. Attention Head Patching and Ablation

**Multiplication Dataset.** Under `mean_shift` corruption, all three heads achieve comparable patching restoration: Head 0 peaks at layer 13 ($0.228\sigma$), Head 1 at layer 12 ($0.196\sigma$), and Head 2 at layer 6 ($0.228\sigma$); corruption parameters are in Appendix B.3. Ablation reveals a sharp asymmetry (Figure 2): Head 2 peaks at layer 0 at $0.076\sigma$, roughly $5\times$ larger than any other head-layer combination, while Heads 0 and 1 show small effects of $0.015\sigma$ and $0.016\sigma$ at layers 12–13.

**Head 2's distinctive ablation profile.** Head 2's ablation peaks at L0 while its patching peaks at L6: the layer of greatest *necessity* differs from the layer of greatest *restora-*

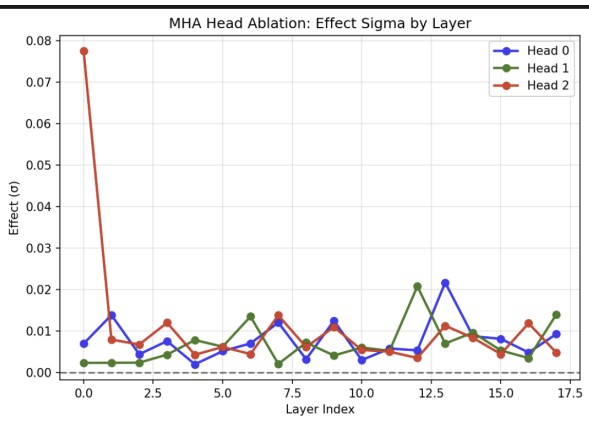

*Figure 2.* MHA attention head ablation, Multiplication Dataset ($n = 512$). Head 2 ablation is largest at layer 0; Heads 0 and 1 peak at layers 12–13.

*bility*. Its attention at L0 is concentrated (entropy 0.22/0.24 on Mult./Pair-50), matching Head 0 (0.22 on both) — yet only Head 2 shows a large ablation effect there, confirming that selective attention is not sufficient for causal necessity.

**Attention entropy confirms targeted computation.** For each head we compute Shannon entropy per query position per sample, average over samples *after* taking entropy (averaging attention weights first would systematically overestimate entropy, by Jensen's inequality), then normalise by $\log F$ (the entropy of a uniform distribution over $F$ feature blocks) so that values are comparable across datasets with different $F$ (full derivation in Appendix E). Lower normalised entropy means attention is more concentrated on fewer feature blocks. Figure 3 shows normalised attention entropy across both datasets; Head 2 has the lowest entropy at L6 (0.21 on both datasets) and at L13 on Pairwise-50 (0.31), coinciding with its largest patching deviations. Heads 0 and 1 are broadly distributed (entropy >0.6) at most layers; Head 0 is co-selective with Head 2 at L0 (0.22 on both datasets) yet has near-zero ablation effect there.

**Heads 0 and 1 as late computation heads.** Heads 0 and 1 exhibit symmetric patching and ablation profiles peaking at layers 12–13 on Multiplication, indicating that their contributions are both necessary and restorable at those depths; a mechanistic interpretation is in Appendix B.3.[1]

**Results: Pairwise-50 Dataset.** Table 1 summarises ablation and entropy results; we use `mean_shift` corruption throughout (details and corruption-mode analysis in Appendix B.3). Figure 4 shows patching results: Head 0 achieves 13.2% gap recovery at layer 5 (positive signed);

---

[1]Head 2's layer-0 ablation effect is stable across sample sizes ($0.078\sigma$ at $n$=64 versus $0.076\sigma$ at $n$=512), confirming this peak is not a single-batch artefact.

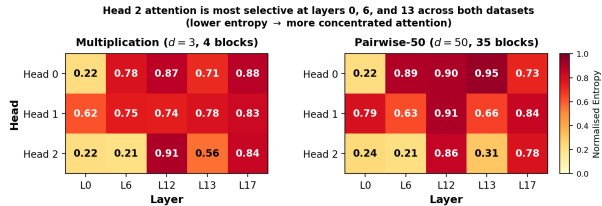

*Figure 3.* Normalised attention entropy per head at five key layers, computed per sample then averaged (see Appendix E). Lower entropy indicates more concentrated attention. Head 2 has the lowest entropy at layer 6 in both datasets (0.21 on both) and additionally at layer 13 on Pairwise-50 (0.31). Heads 0 and 1 maintain higher entropy (>0.6) at most layers. Head 0 at layer 0 is co-selective with Head 2 (entropy 0.22 on both datasets) but has near-zero ablation effect there, demonstrating that attentional selectivity does not imply causal necessity.

*Table 1.* Head ablation effects ($\sigma$) and minimum attention entropy across both datasets. Subscripts denote layer. Patching results (which depend on corruption mode and gap magnitude) are reported in the body text and Figures 4 and 5.

| Head | Ablation ($\sigma$) | | Entropy min | |
|------|------|------|------|------|
| | Mult. | Pair-50 | Mult. | Pair-50 |
| H0 | $0.015_{13}$ | $0.023_{15}$ | $0.22_0$ | $0.22_0$ |
| H1 | $0.016_{12}$ | $0.035_{17}$ | $0.61_0$ | $0.63_6$ |
| **H2** | $\mathbf{0.076_0}$ | $\mathbf{0.074_{16}}$ | $\mathbf{0.21_6}$ | $\mathbf{0.21_6}$ |

Heads 1 and 2 reach their largest *unsigned* deviations at layer 17 (25.5%) and layer 13 (18.5%), both negative signed.

**Head 2's peak ablation magnitude is consistent across tasks; the peak layer is not.** Head 2 peaks at $0.076\sigma$ (L0, Multiplication) and $0.074\sigma$ (L16, Pairwise-50): consistent magnitude but substantially different depth, tentatively attributed to task complexity; two datasets cannot establish this as a general property. Head 1 at layer 17 and Head 2 at layer 13 show negative signed recovery despite large unsigned deviations: substituting clean activations disrupts rather than restores the corrupted computation, consistent with forward-pass divergence by those depths; Head 2's L13 entropy minimum still identifies L13 as computationally active. The same interference pattern is observed for the label block on Multiplication.

## 3. Discussion

**Causal localization.** Head 2's peak ablation magnitude is consistent across tasks ($0.074$–$0.076\sigma$) but its peak layer shifts from L0 to L16; entropy minima and patching deviations converge on the same computationally active layers on each task. Heads 0 and 1 show symmetric late-layer profiles on Multiplication; on Pairwise-50, Head 1 retains this symmetry but Head 0 does not (patching peaks at L5, ablation at L15) — whether these asymmetries scale with task complexity cannot be established from two datasets.

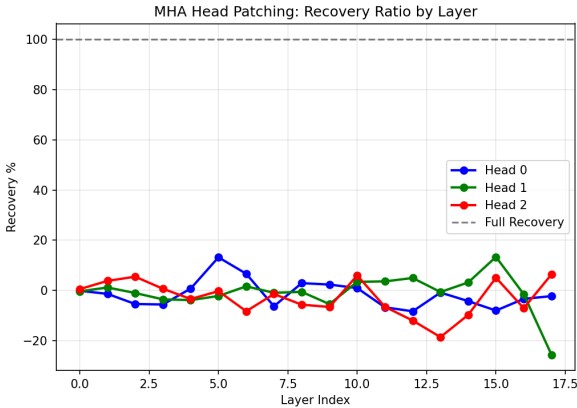

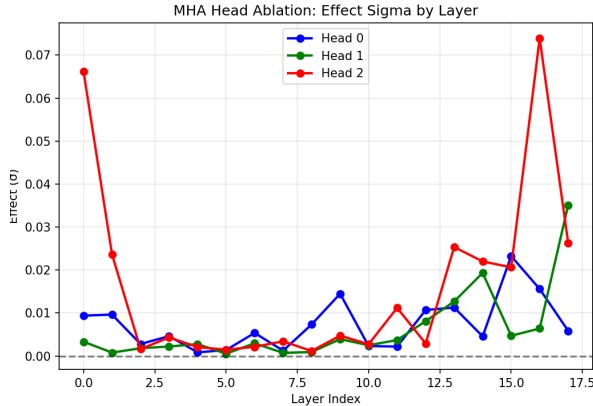

*Figure 4.* MHA head patching recovery ratio (signed), Pairwise-50, `mean_shift` (features 0–9, strength 3.0, $n = 512$). Head 0 achieves 13.2% positive recovery at layer 5. Heads 1 and 2 reach their largest absolute deviations at layer 17 ($|-25.5\%|$, unsigned) and layer 13 ($|-18.5\%|$, unsigned); these are negative-signed recoveries, meaning the patch actively disrupts rather than restores the corrupted computation at those depths (explained below). Left panel (absolute restoration) omitted as a constant rescaling of the ratio.

*Figure 5.* MHA head ablation effect ($\sigma$), Pairwise-50 ($n = 512$). Head 2's peak is at layer 16 ($0.074\sigma$), with a secondary peak at layer 0 ($0.066\sigma$). Heads 0 and 1 show moderate late-layer effects.

**Steerability.** Activation steering experiments (Appendix G) show that contrastive directions do not generalise across samples: a direction computed on a held-out train split produces near-zero MSE improvement on a test split across all hook sites tested. We attribute this to a structural property of pure ICL architectures: unlike LLMs, where few-shot ICL is mediated by function vector heads that produce stable, transferable task representations (Todd et al., 2024; Yin & Steinhardt, 2025; Hendel et al., 2023), TabPFN encodes task relationships entirely through context-dependent attention, leaving no injectable task direction.

**Limitations and future work.** A further limitation concerns our ablation metric: effect-$\sigma$ is measured relative to the model's own unablated prediction on each sample, not as degradation in MSE or $R^2$ against ground-truth labels. A head can therefore register a large ablation effect without this necessarily translating into a comparable loss of predictive accuracy; establishing that link is left to future work. The primary limitation is scope: two synthetic datasets are insufficient to establish the observed head classes as general properties of TabPFN-2.5. The closest cross-dataset evidence is Head 2's peak ablation magnitude ($0.074$–$0.076\sigma$ on both datasets), but the peak layer differs (L0 vs L16); whether this reflects a genuine task-complexity-dependent depth shift or an artefact of these particular tasks requires extending to a broader family of synthetic functions — sinusoidal, polynomial, exponential. Direct attention weight visualisation at L6 and L13 would reveal *which* feature-block pairs Head 2 attends to at its entropy minima, sharpening

the mechanistic account beyond the per-layer selectivity evidence we report. Extending to real-world datasets and classification tasks is an important longer-term direction.

## 4. Conclusion

We presented a causal mechanistic analysis of TabPFN-2.5's feature-wise attention module across two synthetic datasets. Activation ablation at the attention head level identifies two functional head classes: Head 2, whose ablation effect is the largest of any head on both datasets ($0.076\sigma$ at L0 on Multiplication, $0.074\sigma$ at L16 on Pairwise-50), and Heads 0 and 1, which show symmetric patching and ablation profiles at late layers on Multiplication and a partial version of this pattern on Pairwise-50. Head 2's peak ablation *magnitude* is comparable across tasks differing eight-fold in dimensionality, but the peak *layer* differs substantially (L0 to L16); we tentatively attribute this to task complexity, but two datasets cannot establish the depth shift as a general property. Patching under `mean_shift` corruption and attention entropy minima provide converging evidence that L6 (Multiplication) and L13 (Pairwise-50) are layers where Head 2 performs targeted, selective operations.

On steerability, contrastive activation steering fails to transfer to held-out samples — a result we attribute to a structural property of pure ICL architectures: TabPFN encodes task relationships through context-dependent attention compositions rather than the fixed, extractable task vectors that make steering tractable in LLMs (Todd et al., 2024; Yin & Steinhardt, 2025). This is a preliminary finding on a single task, but it identifies a meaningful architectural boundary for steering-based interpretability methods.

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

## A. Experimental Hyperparameters

All experiments use a fixed random seed of 42 unless otherwise noted. Tables 2 and 3 list the key experimental hyperparameters for patching and steering experiments respectively.

*Table 2.* Patching experimental hyperparameters.

| Parameter (default) | Description |
| --- | --- |
| `noise_std` (1.0) | Noise standard deviation ($\geq 0$). |
| `corruption_strength` (1.0) | Corruption strength scalar ($\geq 0$). |
| `seed` (42) | Random seed. |
| `n_samples` (1000) | Dataset size ($\geq 10$). |
| `test_size` (0.5) | Test split proportion, $(0, 1)$. |

*Table 3.* Activation steering hyperparameters.

| Parameter (default) | Description |
| --- | --- |
| `seed` (42) | Random seed |
| `n_samples` (1000) | Dataset size |
| `test_size` (0.5) | Test split proportion |
| steering strength (1.0) | Scaling factor $\alpha$ applied to the direction vector |

## B. Patching and Ablation: Supplementary Results

### B.1. Patching Setup

In activation patching, a clean run $x^{\text{clean}}$ and a corrupted run $x^{\text{corr}}$ differ in a controlled way; the clean activation replaces the corrupted one at site $S$ and the remaining forward pass runs with this patched state:

$$h_{\ell,S}^{\text{patched}}(x^{\text{corr}}) \leftarrow h_{\ell,S}(x^{\text{clean}}).$$

For a scalar regression output $\hat{y}(\cdot)$ we report a normalised restoration score:

$$\text{Restore}(\ell, S) = \frac{\hat{y}(x_{\ell,S}^{\text{patched}}) - \hat{y}(x^{\text{corr}})}{\hat{y}(x^{\text{clean}}) - \hat{y}(x^{\text{corr}})}.$$

### B.2. Layer-Level Patching

For layer-level patching the intervention site is the full residual-stream representation after layer $\ell$:

$$\tilde{H}_\ell = H_\ell^{\text{clean}}.$$

This aggregates all heads and MLP computations and measures the total information carried at each depth. Figure 6 shows $\approx 100\%$ recovery at every layer on the Multiplication Dataset, consistent with a highly distributed representation.

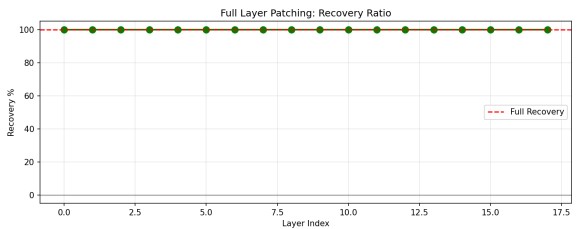

*Figure 6.* Full-layer patching recovery ratio, Multiplication Dataset. ≈100% recovery at every layer.

### B.3. Feature-Block Patching

Feature-block patching replaces one feature-block position $b^*$ in the post-$W_O$ output tensor $A_\ell \in \mathbb{R}^{B \times N \times F \times k}$ (Equation (1)):

$$\tilde{A}_\ell[:, :, b^*, :] = A_\ell^{\text{clean}}[:, :, b^*, :]. \tag{1}$$

This operates on the post-$W_O$ output of `self_attn_between_features` and is coarser than head-level patching: it captures the aggregate contribution of all three heads to that block after recombination. The label block ablation (108% effect ratio at layer 5, see below) confirms `self_attn_between_features` is on the critical computational path; feature-block patching examines which block positions carry that computation.

**Multiplication Dataset** ($n = 512$). For $y = a \cdot b + c$ ($d = 3$), the four blocks are $a$ (index 0), $b$ (index 1, corrupted), $c$ (index 2), and the label (index 3).

Block $b$ (index 1, corrupted) carries the dominant causal signal in early layers (0–4), exceeding 100% recovery: directly patching the corrupted input is the strongest single-block intervention. Block $a$ contributes at $\sim 55\%$, consistent with its role as the non-corrupted partner in $a \cdot b$. Block $c$ is negligible throughout. The label block shows near-zero recovery early, then rises to $\approx +100\%$ at layer 17: by the final layers, the label token's representation alone carries sufficient information to recover the prediction.

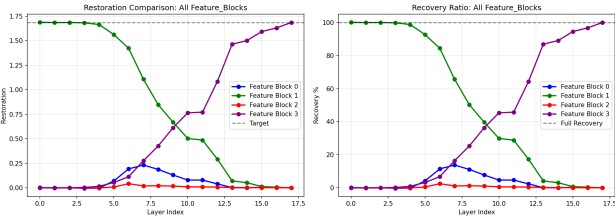

*Figure 7.* Feature-block patching on the Multiplication Dataset ($n = 512$). *Left:* absolute restoration per block. *Right:* recovery ratio (%). Block 1 (green) dominates early layers before handing off to Block 3 (purple) by layer 13.

**Pairwise-50 Dataset** ($n = 512$). All feature blocks produce near-zero recovery across all 18 layers, consistent with

$y = S^2$ requiring global aggregation across all 50 features before squaring. Single-block patching can localise computation only when the interaction is token-local ($a \cdot b$); it cannot for an inherently global computation.

**Corruption parameters and the failure of `gaussian_replace` on Pairwise-50.** For Pairwise-50 patching, `mean_shift` shifts features 0–9 by $+3.0$, producing a signed gap of $-1.81$ — a clean directional signal that supports per-head patching analysis. We initially also tested `gaussian_replace` (corrupting 25 features with strength 1.0 or 3.0), but found this corruption mode produces a near-zero signed mean gap for $y{=}S^2$ due to sign cancellation in $S = \sum_i x_i$ ($\approx -0.05$ across both strengths). At $n = 64$ all heads are at the noise floor ($\approx 0.0003\sigma$); at $n = 512$, gap-normalised recovery percentages can appear large, but per-sample validation reveals these "peaks" are unstable across random seeds and show no consistent directional restoration on the samples that contribute to them. We therefore report only `mean_shift` results in the main body and treat `gaussian_replace` on Pairwise-50 as uninformative for per-head patching. On Multiplication, `gaussian_replace` (feature $b$, strength 1.0) does produce a measurable directional gap, but to maintain a single corruption regime across the head-level analysis we report `mean_shift` numbers throughout the main body.

### B.4. Feature-Block Causal Ablation

Ablation tests necessity: for each feature block $b$ at layer $l$:

$$\text{effect}_{b,l} = y_{\text{normal}} - y_{\text{ablated}}, \quad \text{ratio}_{b,l} = \frac{\text{effect}_{b,l}}{|y_{\text{normal}}|}. \tag{2}$$

Table 4 summarizes results on the Multiplication Dataset.

| Block | Feature | $y_{\text{normal}}$ | $y_{\text{ablated}}$ | Max Effect | Effect Ratio | Best Layer |
|---|---|---|---|---|---|---|
| 0 | $a$ | 4.117 | 2.591 | 1.53 | 37% | 3 |
| 1 | $b$ (corrupted) | 4.117 | 2.508 | 1.61 | 39% | 2 |
| 2 | $c$ | 4.117 | 4.074 | 0.04 | 1% | 5 |
| 3 | label | 4.117 | $-0.32^\dagger$ | 4.44 | 108% | 5 |

*Table 4.* Ablation effects per feature block, $y = a \cdot b + c$. Max Effect $= y_{\text{normal}} - y_{\text{ablated}}$; Effect Ratio $= $ Max Effect$/|y_{\text{normal}}|$. $^\dagger$The 108% effect ratio is measured at layer 5, where the ablated output changes sign, while the 97% reduction is measured at the layer with minimal ablated output; these quantities therefore come from different layers.

The label block (index 3) is the model's most important component. Removing it reduces the prediction by up to 97%, showing that the model relies heavily on it. At layer 5, the measured effect exceeds 100% because the ablated prediction changes sign, making the ratio unstable rather than indicating a stronger effect.

The label block is the output-read position whose feature-attention representation aggregates information

from all feature blocks; its essentiality confirms `self_attn_between_features` is on the critical computational path rather than computation routing through sample-wise attention.

**Blocks** $a$ **and** $b$ contribute moderately ($\approx$40%) and are most influential in early layers (2–3). Block $c$ is dispensable (1% effect), consistent with its purely additive role.

## C. MHA Formulation

The feature-wise self-attention module computes:

$$\text{head}_h = \text{softmax}\left(\frac{XW_Q^{(h)}(XW_K^{(h)})^\top}{\sqrt{d_h}}\right)XW_V^{(h)},$$
(3)

$$\text{MHA}(X) = \text{Concat}(\text{head}_1, \ldots, \text{head}_H)W_O, \quad (4)$$

where $W_Q^{(h)}, W_K^{(h)}, W_V^{(h)} \in \mathbb{R}^{k \times d_h}$ are per-head projection matrices and $W_O \in \mathbb{R}^{Hd_h \times k}$ is the output projection. For TabPFN-2.5: $H=3$, $d_h=64$, $d_{\text{model}}=192$. This decomposition exposes the two intervention granularities described in Section 2.4: feature-block level (patching the post-projection output $A_\ell$) and attention head level (patching per-head outputs $\hat{V}_\ell$ before $W_O$).

## D. Feature Block Construction

All experiments use `TabPFNRegressor` from the `tabpfn==6.3.1` package; TabPFN-2.5 model weights are released under a non-commercial license, while the surrounding package code is Apache 2.0 (Grinsztajn et al., 2025).

The `TabPFNRegressor` default pipeline expands $d$ raw features to $d' = 2d + 1$ preprocessed features: a quantile transform with `append_original=True` doubles the count to $2d$, and one fingerprint feature is appended (Prior Labs, 2025c). The transformer then groups every three consecutive features into a single token and appends one label token (Prior Labs, 2025b), giving $\lceil d'/3 \rceil + 1$ feature blocks per sample. Note that `features_per_group= 3` is specific to the TabPFN-2.5 checkpoint; the TabPFNv2 checkpoint uses a group size of 2 (Ye et al., 2025).

| Dataset | $d$ | $d' = 2d+1$ | $\lceil d'/3 \rceil$ | +label = blocks |
|---|---|---|---|---|
| Multiplication | 3 | 7 | 3 | **4** |
| Pairwise-50 | 50 | 101 | 34 | **35** |

*Table 5.* Feature block counts per dataset. Column 4 is the number of feature groups $\lceil d'/3 \rceil$; adding one label token gives the total block count.

## E. Attention Entropy Computation

For a given head $h$ at layer $\ell$, the feature-wise self-attention module produces an attention tensor of shape [batch $\times$ $N, F, F$] after the softmax, where $F = \lceil d'/3 \rceil + 1$ is the number of feature blocks and $N$ is the number of samples in context. We extract this tensor across $B$ evaluation samples, giving $A \in \mathbb{R}^{B \times F \times F}$ (one matrix per sample), where each row $A[b, q, :]$ is a probability distribution over keys summing to 1.

**Entropy per query per sample.** For each sample $b$ and query position $q$, we compute the Shannon entropy:

$$H_{b,q} = -\sum_{k=1}^{F} A[b,q,k] \log(A[b,q,k] + \varepsilon), \quad \varepsilon = 10^{-12}.$$
(5)

**Averaging.** We average $H_{b,q}$ across all query positions and batch samples:

$$\bar{H} = \frac{1}{B \cdot F}\sum_{b=1}^{B}\sum_{q=1}^{F} H_{b,q}.$$
(6)

This quantity equals the empirical conditional entropy $\hat{H}(K \mid Q)$ — the average uncertainty about which key a randomly drawn query attends to.

**Normalisation.** We normalise by the maximum entropy $\log F$ (achieved by a uniform distribution over keys):

$$\tilde{H} = \frac{\bar{H}}{\log F}.$$
(7)

The normalised entropy $\tilde{H} \in [0, 1]$, where 0 indicates fully concentrated attention (a single key receives all weight) and 1 indicates uniform attention across all keys. Normalising by $\log F$ is necessary for comparing entropy values across the two datasets, which have $F = 4$ and $F = 35$ feature blocks respectively — the maximum possible raw entropy differs by a factor of $\log(35)/\log(4) \approx 2.6$.

**Correctness note.** A naive implementation would average the attention matrices across the batch before computing entropy. This is incorrect: since entropy is a concave function, Jensen's inequality gives $H(\mathbb{E}[A]) \geq \mathbb{E}[H(A)]$, so averaging first systematically overestimates entropy. The values reported in Figure 3 and Table 1 are computed by the correct order: entropy per sample, then average.

**Selectivity versus causal necessity.** On both datasets, Head 0 is co-selective with Head 2 at layer 0 (entropy 0.22 on both datasets for Head 0; 0.22 and 0.24 for Head 2 on Multiplication and Pairwise-50 respectively), yet Head 0

has near-zero ablation effect there while Head 2's L0 ablation effect ($0.076\sigma$) is the largest in the experiment. This demonstrates that attentional selectivity at a given layer is necessary but not sufficient for causal necessity: a head can attend selectively without that selective attention being load-bearing for the prediction. We use the alignment between Head 2's entropy minima and its patching deviation peaks as one of several converging signals, not as a standalone identifier of computational sites.

## F. Token-Level Patching

Token-level patching replaces the full activation vector at a single sequence position in the feature-attention module (`self_attn_between_features`) with the corresponding clean-run activation. The patched tensor has shape $[\text{batch}, \text{tokens}, \text{heads}, d_h]$, so patching token index $t$ replaces one slice along the sequence dimension while leaving all other positions unchanged.

The sequence length depends on the number of eval samples: seq_len $= 264 + n_{\text{eval}}$, with the test-label token always occupying the final position (index seq_len $- 1$).

**Multiplication Dataset, $n_{\text{eval}} = 1$.** With a single eval sample, seq_len$= 265$ and the test-label token is at index 264. Sweeping all 265 token positions, token 264 achieves 100% signed fractional recovery — the only token to do so. All other tokens produce negligible restoration. This is a clean sanity check: causal information funnels entirely through the test-label token position when the computation is isolated to a single sample.

**Multiplication Dataset, $n_{\text{eval}} = 64$.** With 64 eval samples, seq_len$= 328$ and the test-label token is at index 327. Sweeping all 328 positions, the best token is index 324 (not the label position), and restoration is small throughout. The valid signed fractional recovery count is zero across all layers, confirming the low-gap regime. This null result is consistent with the head-level finding: when the eval batch is larger, single-token interventions cannot recover the distributed computation.

**Interpretation.** The contrast between the two runs reflects the structure of the task rather than a failure of the method. At $n_{\text{eval}} = 1$ the output is determined by a single test sample and causal leverage is concentrated at the label token; at $n_{\text{eval}} = 64$ the signal is averaged across samples and no single token position carries sufficient causal weight for recovery. Together, the token-level results confirm that the computation is not localizable by sequence position under averaged eval conditions, consistent with the feature-block null results on the Pairwise-50 Dataset (Appendix B.4).

## G. Activation Steering Experiments

Activation steering (Turner et al., 2023; Panickssery et al., 2023) tests whether injecting a contrastive direction vector $\alpha \cdot \delta$ into the residual stream can shift model outputs toward a target concept. We test whether the multiplicative relationship ($y = a \cdot b + c$) is encoded as a steerable linear direction anywhere in TabPFN-2.5's activation space.

**Setup.** We use the full residual stream at Layer 0 as the hook site — the most complete representation of the model's state at the earliest layer, and the site where Head 2's causal necessity is largest. The contrastive direction is $\delta = \text{mean}(X_{\text{mult}}) - \text{mean}(X_{\text{add}})$, where the additive batch sets $b=0$ (reducing $y = a \cdot b + c$ to $y = c$) while keeping $(a, c)$ fixed. To prevent same-sample leakage, $\delta$ is computed on a held-out train split ($N=512$), $\alpha$ is selected on a val split ($N=256$), and results are reported on a separate test split ($N=256$). The generalizable delta has shape $[F, d_{\text{model}}]$ and norm 0.84 after averaging across samples.

**Result.** Injecting $\alpha \cdot \delta$ at Layer 0 produces near-zero effect on the test split across all $\alpha \in [0, 10]$: MSE improvement is $+0.0\%$ and recovery is $-0.8\%$ at the val-selected $\alpha$. Random and shuffled direction controls show identical flat responses, confirming that the null result is not direction-specific. The direction itself is geometrically stable: cosine similarity between $\delta$ computed on the train split and independently on the val split is 0.71, ruling out the explanation that the direction is simply noisy.

**Interpretation.** The direction is consistent but impotent. The per-sample activation differences between multiplicative and additive contexts are large and structured, but when averaged across samples to extract a generalizable $\delta$, the norm collapses $\approx 300\times$. The information that distinguishes multiplicative from additive contexts is sample-specific: it exists in the attention-weighted composition over the particular context set, not as a shared additive component of the activations.

We argue this is a structural property of pure ICL architectures rather than a feature of the task. TabPFN's weights encode only how to learn from context; the task relationship is computed fresh on every forward pass through attention over the in-context training set. This contrasts with LLMs, where few-shot ICL is driven primarily by function vector (FV) heads — specific attention heads that produce compact, transferable task vectors which can be extracted and re-injected to recover ICL behaviour without any in-context demonstrations (Todd et al., 2024; Hendel et al., 2023; Yin & Steinhardt, 2025). It is precisely this FV mechanism that makes contrastive steering transferable in LLMs: the task representation is stable across inputs because it lives in parametric head weights. TabPFN has no equivalent: its ICL

computation is purely relational (Olsson et al., 2022), the task is read entirely from context, and no head produces a context-independent task vector. Contrastive steering, which requires a stable transferable direction, is therefore architecturally mismatched to this setting.

These are preliminary findings on a single task (Multiplication) and a single direction estimator (mean-diff). Whether the null result holds for Pairwise-50 and more expressive estimators (e.g. linear probes, PCA on contrastive pairs) remains an open question and the primary direction for future work on Q2.

