# OpenReview forum: "Where Computation Lives Inside TabPFN: Causal Localisation of Attention Head Function"
_ICML.cc/2026/Workshop/FMSD — FMSD @ ICML 2026 Poster_

### Official Review · Reviewer_vSZC · 2026-05-12
**Where Computation Lives Inside TabPFN: Causal Localisation of Attention Head Function**

**Rating:** 6
**Confidence:** 4

**Review:**

## Summary
The author conducts analysis on the attention behavior across the layers in TabPFN. Experimental results on two synthetic regression datasets show that the attention effect is dominated by a few peak layers, one dominated at earlier layers, another dominated at only beginning and latter layers.
## Strength
1) The problem being investigated is valuable, where it is important for researchers to understand model inference behavior. Results from such analysis also have the potential to guide distillation training, model pruning, and potentially contribute to the model interpretability.

2) The method and problem setup are reasonable and well written.
## Areas for Improvement
1) The evaluation scenario is a little bit limited also as stated in the limitation section of the paper. Having one or a few representative real world datasets could strengthen the claim significantly.

2) A little bit more details of how the “effect” is calculated will improve the paper a lot. At the same time, having more discussion on alternative effect metrics and more comprehensive study conclusions from literature would be helpful. Example works are: [1] section 4.8, and [2].

[1] NormWear, for encoding arbitrary physiological signal data: Luo, Yunfei, et al. "Toward foundation model for multivariate wearable sensing of physiological signals." ACM HEALTH  (2026).

[2] Bao, Anthony, et al. "Universal Redundancies in Time Series Foundation Models." arXiv preprint arXiv:2602.01605 (2026).

## Detailed Comments
The detailed suggestion is stated in section *Areas for Improvement*.
## Justification of Scores
Overall, this work provides an interesting study aiming to inspect the attention behavior within the foundation model for tabular data. The core contributions of the work nicely aligns with the theme of the workshop titled “Foundation Model for Structured Data”.Tthere is space for improvement as stated above which could improve the convincingness of the claims.

---

### Official Review · Reviewer_VrvN · 2026-05-21
**good paper**

**Rating:** 7
**Confidence:** 4

**Review:**

This paper present a causal mechanistic interpretability analysis of tabular foundation model. If focuses on TabPFN-2.5 feature-wize attention. Using a activation patching and ablation across tow synthetic regression tasks, the authors discover a "division of labor" among the attention heads. Specifically they identify a single dominant head (head 2) that remains necessary across the "easy" and "hard" tasks  and its peak computatoin shifts from easy tasks (first later) to complex task (later layers). The authors also show that contrastive activation steering fails to generalize.

Strenghts:

1. Novel approach that applies mechanistic intrepretability to tabular foundation models.
2. Interesting insights from the experiments, namely the dynamic role of head 2 across easy and hard tasks.
3. The failure of activation steering is an interseting finding that should be valueable to the tabular foundation models community.

Areas for improvement:

The main limlitation is the narrow scope of the experimental setup. I think it is an open question whether these results translate to more tasks, including more synthetic data and real-world data.

Justification of score:

This paper is well-written, creative and insightful. Although the experimental evaluatiion is understandably preliminary, the insights are interesting and valueable to the community.  This work is a strong fit for the workshop and I believe it will be of interest to the attendees.

---

### Official Review · Reviewer_St3k · 2026-05-22
**Careful early mechanistic study of TabPFN, with limited but useful evidence**

**Rating:** 7
**Confidence:** 4

**Review:**

### Summary

This paper studies where computation happens inside TabPFN-2.5, focusing on the feature-wise attention module in the regression model. The authors use activation patching, causal ablation, and attention entropy on two synthetic tasks: a small multiplication task,
$y = ab + c$, and a higher-dimensional Pairwise-50 task, $y = (\sum_i x_i)^2$. The main result is that Head 2 appears to be more causally important than the other heads, although the layer where it matters most changes across the two tasks. The paper also tests contrastive activation steering and finds that the learned direction does not transfer well across samples.

### Strengths

- The topic is very relevant to the workshop. TabPFN-style models are important for structured data, but we still understand very little about how they compute internally.
- The paper uses causal tools rather than only attention visualization. Patching and ablation are appropriate methods for this kind of claim.
- The analysis is reasonably careful. I liked that the authors combine patching, ablation, and entropy instead of relying on one signal.
- The Head 2 finding is interesting, especially because low attention entropy alone does not explain causal importance. This is a useful point for interpretability work.
- The negative steering result is also valuable. Even though it is preliminary, it suggests that TabPFN may not have the same kind of stable task-vector directions seen in some language-model ICL work.

### Weaknesses

- The main limitation is scope. The paper only studies two synthetic regression tasks, so it is hard to know how much the conclusions generalize.
- There are no real tabular datasets, classification tasks, mixed feature types, or missing-data settings. These would matter for a broader claim about TabPFN.
- Some interpretations feel a bit stronger than the evidence. In particular, the steering result is interesting, but one mean-difference direction is not enough to conclude that steering is generally mismatched to TabPFN.
- I would have liked more robustness checks across seeds, corruption choices, and additional synthetic functions.

### Suggestions

The paper would be stronger with one or two more tasks that separate dimensionality from interaction complexity. A simple classification example would also help. For the steering part, testing a linear probe or PCA-based direction would make the negative result more convincing. I also think a direct visualization of what Head 2 attends to at the important layers would make the story easier to interpret.

### Justification of Score

I recommend accept. The paper is preliminary, but it is careful, relevant, and useful for a workshop setting. The claims are not fully general yet, but the work opens a good direction for mechanistic interpretability of tabular foundation models and would likely generate productive discussion.